# Thinning for Accelerating the Learning of Point Processes

**Tianbo Li**,   **Yiping Ke**
School of Computer Science and Engineering
Nanyang Technological University, Singapore
`tianbo001@e.ntu.edu.sg`,  `ypke@ntu.edu.sg`

## Abstract

This paper discusses one of the most fundamental issues about point processes that what is the best sampling method for point processes. We propose *thinning* as a downsampling method for accelerating the learning of point processes. We find that the thinning operation preserves the structure of intensity, and is able to estimate parameters with less time and without much loss of accuracy. Theoretical results including intensity, parameter and gradient estimation on a thinned history are presented for point processes with decouplable intensities. A stochastic optimization algorithm based on the thinned gradient is proposed. Experimental results on synthetic and real-world datasets validate the effectiveness of thinning in the tasks of parameter and gradient estimation, as well as stochastic optimization.

## 1 Introduction

Point processes are a powerful statistical tool for modeling event sequences and have drawn massive attention from the machine learning community. Point processes have been widely used in finance [6], neuroscience [8], seismology [28], social network analysis [22] and many other disciplines. Despite their popularity, applications related to point processes are often plagued by the scalability issue. Some state-of-the-art models [37, 35, 33] have a time complexity of $\mathcal{O}(d^2 n^3)$, where $n$ is the number of events and $d$ is the dimension. As the number of events increases, learning such a model would be very time consuming, if not infeasible. This becomes a major obstacle in applying point processes.

A simple strategy to address this problem is to use part of the dataset in the learning. For instance, in mini-batch gradient descent, the gradient is computed at each iteration using a small batch instead of full data. For point processes, however, to find a suitable sampling method is not a easy task, at least not as easily as it might seem. This is due to the special input data — event sequences. First of all, event sequences are *posets*. An inappropriate sampling methods may spoil the order structure of the temporal information. This is especially harmful when the intensity function depends on its history. Second, many models built upon point processes utilize the arrival intervals between two events. Such models are particularly useful as they take into account the interactions between events or nodes. Examples include Hawkes processes and their variants [37, 33, 19]. An improper sampling method may change the lengths of arrival intervals, leading to a poor estimation of model parameters.

A commonly-used approach to the sampling of point processes is *sub-interval sampling* [34, 32]. Sub-interval sampling is a piecewise sampling method, which splits an event sequence into small pieces and learns the model on these sub-intervals. At each iteration, one or several sub-intervals are selected to compute the gradient. This method, however, has an intrinsic limitation: it cannot capture the panoramic view of a point process. Take self-excited event sequences as an example. An important characteristic of such sequences is that events are not evenly distributed across the time

axis, but tend to be clumping in a short period of time. Sub-interval sampling, in this circumstance, is like "a blind man appraising an elephant" — it can only see part of the information at each iteration, prone to a large variance of the gradient.

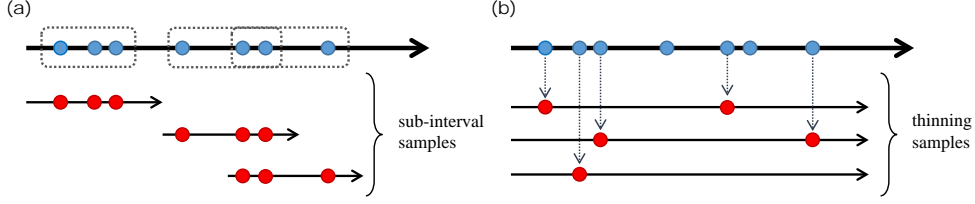

Figure 1: Comparison of sub-interval sampling (a) and thinning sampling (b).

In this paper, we discuss "thinning sampling" as a downsampling method for accelerating the learning of point processes. A comparison between sub-interval and thinning are shown in Figure 1. Conventionally, thinning is a classic technique for simulating point processes [18]. We borrow the idea and adopt it as a downsampling method for fast learning of point processes. It is convenient to implement and able to capture the entire landscape over the observation timeline.

The main contributions of this paper are summarized as follows.

- To the best of our knowledge, we are the first to employ thinning as a downsampling method to accelerate the learning of point processes.
- We present theoretical results for intensity, parameter and gradient estimation on the thinned history of a point process with decouplable intensities. We also apply thinning in stochastic optimization for the learning of point processes.
- Experiments verify that thinning sampling can significantly reduce the model learning time without much loss of accuracy, and achieve the best performance when training a Hawkes process on both synthetic and real-world datasets.

## 2 Point Processes

A point process $N(t)$ can be viewed as a random measure from a probability space $(\Omega, \mathfrak{F}, P)$ onto a simple point measure space $(\mathcal{N}, \mathfrak{B}_{\mathcal{N}})$. We define a point process $N(t)$ as follows.

**Definition 2.1** (Point process)**.** *Let* $t_i$ *be the* $i$*-th arrival time of a point process* $N(t)$ *defined by,* $t_i = \inf_t \{N(t) \geqslant i\}$*. A point process* $N(t)$ *on* $\mathbb{R}^+$ *is defined by* $N(t) = \sum_i \delta_{t_i}(t)$*, where* $\delta_\omega$ *is the Dirac measure at* $\omega$*.*

The "information" available at time $t$ is represented by a sub-$\sigma$-algebra $\mathcal{H}_t = \sigma(N(t) : t \in \mathbb{R}^+)$. The filtration $\mathcal{H} = (\mathcal{H}_t)_{0 \leqslant t < \infty}$ is called the *internal history*. A point process can be characterized by its *intensity function*. It measures the probability that a point will arrive in an infinitesimal period of time given the history up to the current time. Herein we follow the definition of stochastic intensity introduced in [7, 15].

**Definition 2.2** (Stochastic intensity)**.** *Let* $N(t)$ *be an* $\mathcal{H}$*-adapted point process and* $\lambda(t)$ *be a nonnegative* $\mathcal{H}$*-predictable process.* $\lambda(t)$ *is called the* $\mathcal{H}$*-intensity of* $N(t)$*, if for any nonnegative* $\mathcal{H}$*-predictable process* $C(t)$*, the equality below holds,*

$$\mathbb{E}\left[\int_0^\infty C(s)dN(s)\right] = \mathbb{E}\left[\int_0^\infty C(s)\lambda(s)ds\right]. \tag{1}$$

The expectation of $N(t)$ is called the $\mathcal{H}$*-compensator*, which is the cumulative intensity $\Lambda(t) = \int_0^t \lambda(s)ds$. Doob-Meyer decomposition yields that $N(t) - \Lambda(t)$ is an $\mathcal{H}$-adapted martingale. Another important result is that the conditional intensity function uniquely determines the probability structure of a point process [9]. Similar results can be extended to compensators [10, 15].

**M-estimator.** Commonly-used parameter estimation methods for point processes include maximum likelihood estimation (MLE) [22, 33, 37], intensity-based least square estimation (LSE) [3, 21] and counting-based LSE [32]. These methods fall into a wider class called *martingale estimator* (M-estimator) [15, 2]. The gradient $\nabla R(\theta)$, $\theta \in \mathbb{R}^d$ can be expressed as a stochastic integral: $N(t)$,

*(full gradient)*
$$\nabla R(\theta) = \int_0^T H(t; \theta) \left[ dN(t) - \lambda(t; \theta) dt \right].$$
(2)

Here $R(\theta)$ is the loss function, which may be the log-likelihood, or the sum of the squares of the residuals. $\lambda(t; \theta)$ is the intensity function to be estimated. $[0, T]$ is the observation window. $H(t; \theta)$ is a vector-valued function, or more generally, a predictable, boundedly finite, and square integrable process associated with $\lambda(t; \theta)$. Different choices of $H(t; \theta)$ instantiate different estimators: $H(t; \theta) = -\nabla \log \lambda(t; \theta)$ for MLE, $H(t; \theta) = \nabla \lambda(t; \theta)$ for intensity-based LSE, and $H(t; \theta) = 1$ for counting-based LSE. We write $\sum_{i=1} \nabla R(\theta; \omega_i)$, $\omega_i \in \Omega$ as the empirical gradient given realizations $\{\omega_i\}$. Under the true parameter $\theta^*$, $\nabla R(\theta^*)$ is a martingale and its expectation is 0. Gradient descent methods are often used to find $\hat{\theta}$ by solving $\sum_{i=1} \nabla R(\hat{\theta}; \omega_i) = 0$. Note that in this paper, LSE refers to intensity-based LSE, as all the results can be easily extended to counting-based LSE.

## 3 Thinned Point Processes

In this section, we introduce a derivative process called *p-thinned process*. Intuitively, the p-thinned process is obtained from a point process by retaining each point in it with probability p, and dropping with probability $1 - p$. We formally define the *p*-thinned process as follows.

**Definition 3.1** (p-thinned process). *The* p-*thinned process* $N_p(t)$ *associated with a point process* $N(t)$ *(called the ground process) is defined by summing up the Dirac measure on the product space* $\Omega \times \mathcal{K}$:
$$N_p(t) = \sum_{i=1} \delta_{(t_i, B_i=1)}(t),$$
*where* $B_i$*'s are independent Bernoulli distributed random variables with parameter* p.

Alternatively, the p-thinned process can be written in the form of a compound process as $N_p(t) = \sum_{i=1}^{N(t)} B_i$ and its differential can be expressed as $dN_p(t) = B_{N(t)} dN(t)$. In this way, the RiemannStieltjes integral of a stochastic process with respect to a p-thinned process can be defined by,
$$\int_0^T H(t) dN_p(t) = \int_0^T H(t) B_{N(t)} dN(t) = \sum_{i=1}^{N(T)} H(t_i | \mathcal{H}_{t_i-}) B_i.$$

**Two types of histories.** The differential defined above implies that the intensity of thinned process can be written as $\lambda_p(t) = p\lambda(t)$. This relation between the intensities of a thinned process and its ground process is intuitively plausible. The implicit condition, however, is that $\lambda_p(t)$ must be measurable with respect to the full history of $N(t)$ and all the thinning marks $B_i$ prior to the current time t. Such a history can be expressed by the filtration $\mathcal{F} = (\mathcal{F}_t)$, where $\mathcal{F}_t = \mathcal{H}_t \otimes \mathcal{K}_t$ and $\mathcal{K}_t$ is the cylinder $\sigma$-algebra of the markers. This history is called the *full history*, and its corresponding $\mathcal{F}$-intensity is denoted by $\lambda_p^{\mathcal{F}}(t)$. When computing $\lambda_p^{\mathcal{F}}(t)$, we need to take into account all the points prior to t, including those dropped ones.

The other type of history, called *the thinned history*, is the internal history of the thinned process, denoted by $\mathcal{G} = (\mathcal{G}_t)$, where $\mathcal{G}_t = \sigma(N_p(t))$. In computing its $\mathcal{G}$-intensity $\lambda_p^{\mathcal{G}}(t)$, we only need to consider all the retained points of the thinned process.

The following lemma describes the relationship between different intensities.

**Lemma 3.2** (Relationship of intensities). *Let* $\mathcal{F}$ *and* $\mathcal{G}$ *be the full history and thinned history with respect to a* p-*thinned process* $N_p(t)$. *Let* $\mathcal{H}$ *be the internal history of* $N(t)$. *The following equalities hold:*

*(1)* $\lambda_p^{\mathcal{F}}(t) = p\lambda^{\mathcal{H}}(t)$;

*(2)* $\lambda_p^{\mathcal{G}}(t) = p\mathbb{E}\left[\lambda^{\mathcal{H}}(t) | \mathcal{G}\right]$.

Due to space limit, we defer all the proofs to the supplementary material. Lemma 3.2 tells that the intensity of a p-thinned process is a version of the conditional expectation $\mathbb{E}\left[\lambda^{\mathcal{H}}(t)|\mathcal{G}\right]$. Provided the information of the p-thinned process, the p-thinned intensity is an orthogonal projection of that of the ground point process on $\mathcal{L}^2$. Therefore, $1/p\lambda_p^{\mathcal{G}}(t)$ is an unbiased estimation of $\lambda^{\mathcal{H}}(t)$.

**p-thinned and sub-interval gradient.** We define the p-*thinned gradient*, which is the stochastic integral with respect to a p-thinned process:

$$(\text{p-thinned gradient}) \qquad \nabla R_p(\theta) = \frac{1}{p}\int_0^T H_p^{\mathcal{G}}(t;\theta)\left[dN_p(t) - \lambda_p^{\mathcal{G}}(t;\theta)dt\right]. \qquad (3)$$

Here $H_p^{\mathcal{G}}(t;\theta)$ is related to $\lambda_p^{\mathcal{G}}(t;\theta)$.

Sub-interval gradient is defined as follows. Let $\tau_0, \tau_1, \tau_2, \ldots, \tau_{\lceil 1/p\rceil}$ be a partition of $[0, T)$, where $\tau_0 = 0$, and $\tau_{\lceil 1/p\rceil} = T$. We cut the interval into $\lceil 1/p\rceil$ pieces so that the batch size is comparable to that in the thinned gradient. At every step, one interval is selected with probability p. We define the *sub-interval gradient* on $[\tau_i, \tau_{i+1})$ by,

$$(\text{sub-interval gradient}) \qquad \nabla R_\ell(\theta) = \frac{1}{p}\int_0^T I\{t\in[\tau_i,\tau_{i+1})\}H^{\mathcal{F}}(t;\theta)\left[dN(t) - \lambda^{\mathcal{F}}(t;\theta)dt\right],$$

where I is an indicator representing whether the sub-interval is selected or not. Here we consider the full history. It can be easily verified that $\nabla R_\ell(\theta)$ is an unbiased estimation of the full gradient in Eq. (2). The thinned gradient can be used as a estimator of full gradient, which will be illustrated in Section 5. This definition also generalizes the stochastic optimization method proposed in [32], which splits the observation timeline at the arrival of each event.

# 4 Thinning for Parameter Estimation

In this section, we discuss how to estimate the parameter $\theta \in \mathbb{R}^d$ of the intensity function $\lambda(t;\theta)$ given the thinned history $\mathcal{G}$. We first define the notations used.

- $\theta_{\mathcal{H}}^*$: true parameter of $\mathcal{H}$-intensity $\lambda^{\mathcal{H}}(t;\theta)$, such that $\mathbb{E}\nabla R(\theta_{\mathcal{H}}^*) = 0$;
- $\theta_{\mathcal{G}}^*$: true parameter of $\mathcal{G}$-intensity $\lambda_p^{\mathcal{G}}(t;\theta)$, such that $\mathbb{E}\nabla R_p(\theta_{\mathcal{G}}^*) = 0$;
- $\hat{\theta}_{\mathcal{H}}$: estimate of $\theta_{\mathcal{H}}^*$, such that $\sum_i \nabla R(\hat{\theta}_{\mathcal{H}}; \omega_i) = 0$;
- $\hat{\theta}_{\mathcal{G}}$: estimate of $\theta_{\mathcal{G}}^*$, such that $\sum_i \nabla R_p(\hat{\theta}_{\mathcal{G}}; \omega_i') = 0$, where $\omega_i'$ is a realization of the p-thinned process.

The task of parameter estimation on a thinned history is to find $\tilde{\theta}_{\mathcal{H}}$, such that $\mathbb{E}\left[\nabla R(\tilde{\theta}_{\mathcal{H}})|\mathcal{G}\right]$ is close enough to 0. We refer to $\tilde{\theta}_{\mathcal{H}}$ as *the M-estimator on thinned history*. Here the expectation is over the thinning operation. The tilde is used to indicate that $\tilde{\theta}_{\mathcal{H}}$ is a $\mathcal{G}$-measurable estimator for the parameter of $\mathcal{H}$-intensity $\lambda^{\mathcal{H}}(t;\theta)$, whereas $\hat{\theta}_{\mathcal{H}}$, with a hat on it, is $\mathcal{H}$-measurable. A notable result is that M-estimators have asymptotic normality [2], thus we have $\hat{\theta}_{\mathcal{H}} \xrightarrow{P} \theta_{\mathcal{H}}^*$ and $\hat{\theta}_{\mathcal{G}} \xrightarrow{P} \theta_{\mathcal{G}}^*$, as the number of realizations $n \to \infty$.

In the following, we first present a method for parameter estimation of a non-homogeneous Poisson process (NHPP) whose intensity is deterministic. We then derive a theorem that works for a more general type of intensities.

**Lemma 4.1** (Thinning for parameter estimation of NHPP)**.** *Consider an NHPP* $N(t)$ *with deterministic intensity* $\lambda(t;\theta)$, $t > 0$, $\theta \in \mathbb{R}^d$. *If there exists an invertible linear operator* $\mathcal{A}: \mathbb{R}^d \to \mathbb{R}^d$ *satisfying* $\lambda(t; \mathcal{A}\theta) = p\lambda(t;\theta)$, *then the M-estimator on thinned history can be written as* $\tilde{\theta}_{\mathcal{H}} = \mathcal{A}^{-1}\hat{\theta}_{\mathcal{G}}$ *such that* $\mathbb{E}\left[\nabla R(\tilde{\theta}_{\mathcal{H}})|\mathcal{G}\right] \xrightarrow{P} 0$, *as the number of realizations* $n \to \infty$.

**Example** (Parameter estimation for NHPP). Let consider an NHPP with intensity $\lambda(t; a, b, c, d) = a + b\sin(ct+d)$. We can find a diagonal matrix $\mathcal{A} = \text{diag}(p, p, 1, 1)$ such that $\lambda(t; \mathcal{A}(a, b, c, d)) = pa + pb\sin(ct + d) = p\lambda(t; a, b, c, d)$. Thus the parameter given the thinned history can be estimated by $\mathcal{A}^{-1}(\hat{a}, \hat{b}, \hat{c}, \hat{d}) = (1/p\hat{a}, 1/p\hat{b}, \hat{c}, \hat{d})$, where $\hat{a}, \hat{b}, \hat{c}, \hat{d}$ are estimated on the thinned history.

Next, we focus on a more general type of intensities, called *decouplable intensity*. Most commonly-used point processes have decouplable intensities, including NHPPs, linear Hawkes processes, compound Poisson process, etc.

**Definition 4.2** (Decouplable intensity). *An intensity function is said to be decouplable, if it can be written in such a form:*

$$\lambda^{\mathcal{H}}(t; \theta) = g(t; \theta)^{\mathsf{T}} m^{\mathcal{H}}(t). \tag{4}$$

*Here $g(t; \theta)$ is a deterministic vector-valued function that is continuous with respect to $\theta$ and does not contain any information regarding $\mathcal{H}_t$. $m^{\mathcal{H}}(t)$ is an $\mathcal{H}$-predictable vector-valued measure that does not contain any information regarding $\theta$. Particularly, $\lambda^{\mathcal{H}}(t; \theta)$ is said to be linear if $g(t; \theta) = \theta$.*

This category covers a multitude of state-of-the-art models, including Netcodec [30], parametric Hawkes [19], MMEL model [35], Granger causality for Hawkes [33], and the sparse low-rank Hawkes [36]. The next theorem demonstrates a similar result with Lemma 4.1 for decouplable intensities.

**Theorem 4.3** (Thinning for parameter estimation of decouplable intensities). *Consider a point process $N(t)$ with decouplable intensity. If there exist invertible linear operators $\mathcal{A}$ and $\mathcal{B}$ satisfying $\mathcal{B}\mathbb{E}[m^{\mathcal{H}}(t)|\mathcal{G}] = m_p^{\mathcal{G}}(t)$, where $m_p^{\mathcal{G}}(t)$ is the component of thinned intensity $\lambda_p^{\mathcal{G}}(t)$, and $p\mathcal{B}^{-1}g(t; \theta) = g(t; \mathcal{A}\theta)$, then the M-estimator on thinned history can be written as $\tilde{\theta}_{\mathcal{H}} = \mathcal{A}^{-1}\hat{\theta}_{\mathcal{G}}$ such that $\mathbb{E}[\nabla R(\tilde{\theta}_{\mathcal{H}})|\mathcal{G}] \xrightarrow{P} 0$, as the number of realizations $n \to \infty$. Particularly, if $\lambda^{\mathcal{H}}(t; \theta)$ is linear, then $\mathcal{A} = p\mathcal{B}^{-1}$.*

**Example** (Parameter estimation for Hawkes processes). Consider a one-dimensional Hawkes process with intensity $\lambda^{\mathcal{H}}(t; \mu, \alpha) = (\mu, \alpha)^{\mathsf{T}}(1, m^{\mathcal{H}}(t))$, where $m^{\mathcal{H}}(t) = \sum_{i=1} \phi(t - t_i)$. From the fact that $\mathbb{E}[m^{\mathcal{H}}(t)|\mathcal{G}] = 1/p\, m_p^{\mathcal{G}}(t)$, we obtain $\mathcal{B} = \text{diag}(1, p)$. Thus Theorem 4.3 yields $\mathcal{A} = p\mathcal{B}^{-1} = \text{diag}(p, 1)$ and consequently $\mu$ and $\alpha$ can be estimated by $p\hat{\mu}$ and $\hat{\alpha}$, where $\hat{\mu}$ and $\hat{\alpha}$ are estimated on the thinned history. Similar results can be obtained on multi-dimensional linear Hawkes processes. This result reveals that the thinning operation does not change the endogenous triggering pattern in linear Hawkes processes.

**Remark** (Parameter estimation for multi-dimensional Hawkes processes). The thinning estimator is also valid for multi-dimensional Hawkes processes. Consider the $i$-th dimension of an $m$-dimensional Hawkes process. Its intensity function can be written as $\lambda_i^{\mathcal{H}}(t; \mu_i, \alpha_{i1}, \cdots, \alpha_{im}) = (\mu_i, \alpha_{i1}, \cdots, \alpha_{im})^{\mathsf{T}}(1, m_1^{\mathcal{H}}(t), \cdots, m_m^{\mathcal{H}}(t))$, which complies with the definition of decouplable intensity. Theorem 4.3 again yields a thinning estimator with the linear operator $\mathcal{A} = \text{diag}(p, 1, ..., 1)$.

## 5 Thinning for Gradient Estimation and Stochastic Optimization

So far we have discussed how to estimate the parameter given the thinned history. In fact, the gradient at any $\theta$ can also be recovered without knowing all the information about a point process. The following theorem describes the gradient estimation on the thinned history for decoupleable intensity.

**Theorem 5.1** (Thinning for gradient estimation). *Let $N(t)$ be a point process with decouplable intensity $\lambda^{\mathcal{H}}(t; \theta) = g(t; \theta)^{\mathsf{T}} m^{\mathcal{H}}(t)$ in Eq. (4). If there exist invertible linear operators $\mathcal{A}$ and $\mathcal{B}$ satisfying $\mathcal{B}\mathbb{E}[m^{\mathcal{H}}(t)|\mathcal{G}] = m_p^{\mathcal{G}}(t)$, where $m_p^{\mathcal{G}}(t)$ is the component of thinned intensity $\lambda_p^{\mathcal{G}}(t)$, and $p\mathcal{B}^{-1}g(t; \theta) = g(t; \mathcal{A}\theta)$, then*

*(1) $\mathbb{E}[\nabla R(\theta)|\mathcal{G}] \leqslant 1/p\mathcal{A}^{-1}\nabla R_p(\mathcal{A}\theta)$, for R is LSE;*

*(2) $\mathbb{E}[\nabla R(\theta)|\mathcal{G}] \leqslant \mathcal{A}^{-1}\nabla R_p(\mathcal{A}\theta)$, for R is MLE.*

*Particularly, if the intensity is deterministic, i.e., $m^{\mathcal{H}}(t) = 1$, both equalities hold.*

**Remark**. The thinned gradient can be transformed to a larger estimation of the full gradient, and an unbiased estimation for deterministic intensity. More specifically, the gradient estimation is unbiased if and only if $\mathbb{E}[\mathcal{H}(t; \theta)\lambda(t; \theta)] = \mathbb{E}\mathcal{H}(t; \theta)\mathbb{E}\lambda(t; \theta)$, as shown in the proof of Theorem 5.1. Here $\mathcal{H}$ usually depends on the intensity function $\lambda(t; \theta)$, such as MLE estimator has $\mathcal{H}(t; \theta) = -\nabla \log \lambda(t; \theta)$. The condition may not hold under such circumstances. For stochastic

intensities, the thinned gradient may be biased, yielding an estimation larger than the ground truth. Some empirical results on Hawkes processes are shown in Figure 3. The next theorem shows that the thinned gradient has a smaller variance compared with the sub-interval gradient.

**Theorem 5.2** (Variance comparison)**.** *Let* $\nabla \tilde{R}^{\mathcal{G}}(\theta)$ *and* $\nabla R_{\ell}(\theta)$ *be the* $p$-*thinned and sub-interval gradient at* $\theta$, *where* $\nabla \tilde{R}^{\mathcal{G}}(\theta) = 1/p\mathcal{A}^{-1}\nabla R_p(\mathcal{A}\theta)$ *for LSE and* $\nabla \tilde{R}^{\mathcal{G}}(\theta) = \mathcal{A}^{-1}\nabla R_p(\mathcal{A}\theta)$ *for MLE. The variance of the* $p$-*thinned gradient is no greater than that of the sub-interval gradient, i.e.,*

$$\mathbb{V}\Big[\nabla \tilde{R}^{\mathcal{G}}(\theta)\Big] \leqslant \mathbb{V}\Big[\nabla R_{\ell}(\theta)\Big].$$

**Remark**. A Chebyshev error bound can be easily obtained, as a result of Theorems 5.1 and 5.2:

$$\mathbb{P}\Big(\big|\nabla \tilde{R}^{\mathcal{G}} - \mathbb{E}\nabla \tilde{R}^{\mathcal{G}}(\theta)\big| > \epsilon\Big) \leqslant \frac{\mathbb{V}\Big[\nabla \tilde{R}^{\mathcal{G}}(\theta)\Big]}{\epsilon^2} \leqslant \frac{\mathbb{V}\Big[\nabla R_{\ell}(\theta)\Big]}{\epsilon^2} = \frac{\frac{1-p}{p}\Big[\mathbb{E}\nabla R(\theta)\Big]^2 + \frac{1}{p}\mathbb{V}\big[\nabla R(\theta)\big]}{\epsilon^2},$$

for any $\epsilon > 0$. Since $\nabla R(\theta)$ is a martingale integral (Eq. 2), we have $\mathbb{E}\nabla R(\theta) \to 0$, as the number of realizations increases. Hence, the left-hand side probability is bounded by $\mathcal{O}(\epsilon^{-2}p^{-1}\mathbb{V}\big[\nabla R(\theta)\big])$, which shows that the gradient estimation of deterministic intensities will not be far from its true one, if the number of realizations is sufficiently large. Unfortunately, the result does not apply to stochastic intensities. Nonetheless, its effectiveness on stochastic intensities is empirically validated on real datasets with Hawkes processes in our experiments (See Figure 4).

**Thinning for stochastic optimization.** We have shown that thinning can be used for estimating parameters and gradients with less data.This inspires us to employ it to stochastic optimization. We propose a novel Thinning-SGD (TSGD) method for learning a point process with a parametric intensity function, as shown in Algorithm 1. At each iteration, a thinned dataset is used for computing the gradient. Compared with sub-interval variance, thinned gradient has a smaller variance, so that the convergence curve may have less fluctuations and find a path to the optimal solution faster. Thinning is also applicable to other gradient-based optimization algorithms such as Adam [16].

---

**Algorithm 1:** TSGD: Thinning Stochastic Gradient Descent

---

**Input** :Event sequences $\{t_i\}$, learning rate $\alpha$, thinning size $p$, convergence criterion, the objective function of a parametric point process model $R(\theta)$.
**Output :**Optimal parameter $\theta^*$.
1 Initialize $\theta$;
2 Find $\mathcal{A}$ according to Theorem 4.3;
3 **repeat**
4      Sample a $p$-thinning batch $t_i'$ from one of the sequences $t_i$;
5      Compute the thinned gradient $\tilde{R}^{\mathcal{G}}(\theta)$, where $\tilde{R}^{\mathcal{G}}(\theta)$ is defined in Theorem 5.2;
6      $\theta \leftarrow \theta - \alpha \tilde{R}^{\mathcal{G}}(\theta)$ ;
7 **until** *Convergence criterion is satisfied*;

---

## 6 Related Work

**Learning of parametric point processes.** Parametric point processes are the most conventional and popular method in the study of point processes. For example, [37] designs an algorithm ADM4 for learning the parameter representing the hidden network of social influences. [19] parameterizes the infectivity parameter in Hawkes processes and employs the technique of ADMM for parameter estimation. [33] proposes a learning algorithm combining MLE with a sparse-group-lasso regularizer to learn the so-called "Granger causality graph". All these models are decouplable, therefore thinning is applicable to the learning of them.

**Learning of non-parametric point processes.** There has been an increasing amount of studies on non-parametric point processes and their learning algorithms in recent years. Isotonic Hawkes process [31] is an interesting and representative work among them, which combines isotonic regression and Hawkes processes. [1] proposes a algorithm to learn the infectivity matrix without any parametric modeling and estimation of the kernels. Another category of non-parametric models related

to point processes is Bayesian non-parametric models, such as [4], [12] and [25]. Besides, some explorations of combining point processes and deep neural networks are emerging. Some typical works include [11], [26], and [20].

**Acceleration for the learning of point processes.** [17] proposes a method of low rank approximation of the kernel matrix for large-scale datasets. The online learning algorithm for Hawkes [34] discretizes the time axis into small intervals for learning the triggering kernels. [13] designs a hardware acceleration method for MLE of Hawkes processes. A recent work [32] introduces an stochastic optimization method for Hawkes processes. Unfortunately, none of existing works considers thinning as a sampling methods to reduce the time complexity.

**Thinning for point processes.** The thinning operation of point processes has been discussed mainly in the statistics community. Thinning is first used for the simulation of point processes [18, 27]. Some limit results have been proposed [14, 29, 5], among which the property of Cox process approximation is often mentioned [10]. However, most, if not all, of these asymptotic results investigate the behavior of a thinned process as the thinning level $p \to 0$, which does not serve our purpose.

## 7 Experiments

In this section, we assess the performance of our proposed thinning sampling in three tasks: parameter estimation, gradient estimation, and stochastic optimization. All the experiments were conducted on a server with Intel Xeon CPU E5-2680 (2.80GHz) and 250GB RAM.

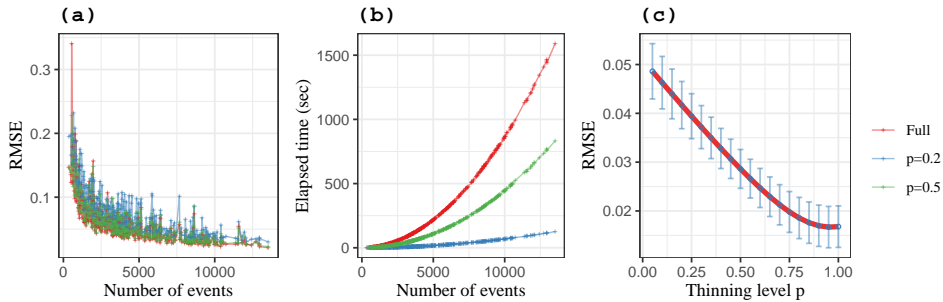

Figure 2: Parameter estimation on a 10-dimensional linear Hawkes process with LSE. (a): the RMSE of estimated parameters. (b): trainning time. (c): RMSE v.s. thinning level $p$.

**Parameter estimation.** We conduct two experiments for this task on synthetic datasets. The first experiment is to test thinning on Hawkes processes. We simulate 100 sequences of 10-dimensional linear Hawkes processes and use different number of events for training. The longest sequence has around 14k events. The parameters of the process are randomly generated from a uniform distribution. For each dataset, we perform LSE with different histories: full data and p-thinned data with $p = 0.2$ and $p = 0.5$.

The results are shown in Figure 2. We can see that as the number of events training increases, the error (measured by RMSE) in parameter estimation decreases, at the cost of longer running time. A

Table 1: Parameter estimation on state-of-the-art models.

| | Model | RMSE/Accuracy | Training time (s) |
|---|---|---|---|
| MMEL [37] | Full | 0.0568 (0.0013) | 38.03 (4.19) |
| | Thinned (p=0.5) | 0.0569 (0.0012) | 8.68 (1.06) |
| | Thinned (p=0.2) | 0.0570 (0.0012) | 3.94 (0.47) |
| Granger Causality for Hawkes [33] | Full | 0.0161 (0.0078) | 229.56 (17.87) |
| | Thinned (p=0.5) | 0.0163 (0.0022) | 65.68 (4.67) |
| | Thinned (p=0.2) | 0.0167 (0.0010) | 3.96 (1.80) |
| Sparse Low-rank Hawkes [35] | Full | 97.46% (0.0133) | 73.76 (42.24) |
| | Thinned (p=0.5) | 97.60% (0.0166) | 27.45 (17.51) |
| | Thinned (p=0.2) | 96.63% (0.0243) | 4.51 (2.65) |

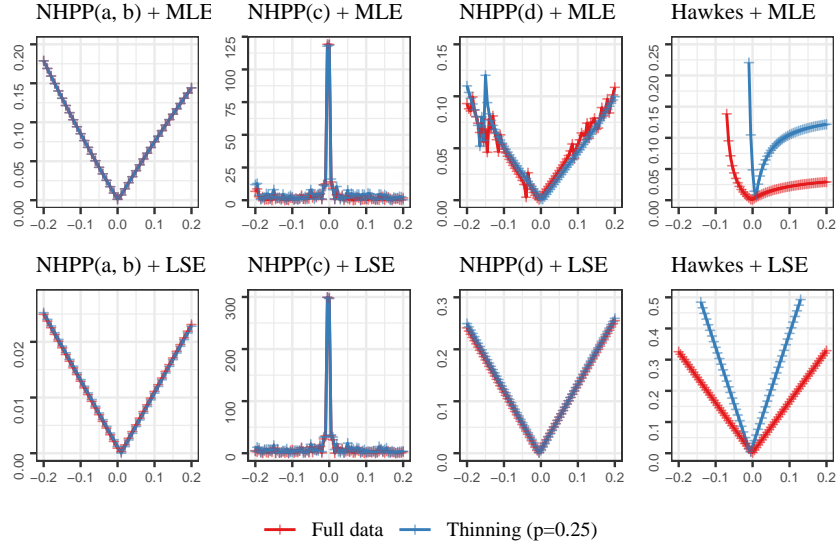

Figure 3: Gradient estimation for an NHPP and a linear Hawkes process using MLE and LSE. X-axes represent the RMSE of the parameters, and Y-axes the $l_2$-norm of gradient with corresponding parameters.

larger p value yields better estimations but also runs slower. When the number of events is large enough, the estimation with 0.2-thinning is as accurate as that with full data, but runs an order of magnitude faster. For a dataset of 14k events, 0.2-thinning only took 2 minutes, whereas the training on full data took 26.5 minutes, and the decrease of RMSE is less than 0.01. Figure 2 (c) shows that RMSE decreases as the thinning level p increases.

The second experiment is to test thinning for learning various state-of-the-art models: MMEL [37], Granger Causality for Hawkes [33] and Sparse Low-rank Hawkes [35]. We generate 30 sequences for each model and perform parameter estimation on different histories. The averages and standard deviations of the quality metric and training time are presented in Table 1. We use RMSE as the metric for MMEL and Granger Causality, and the accuracy of non-zero entries in the adjacency matrix for Sparse Low-rank Hawkes. It can be seen that thinning significantly reduces the training time of all models without compromising much estimation quality.

**Gradient estimation.** We consider two types of point process: a non-homogeneous point process with deterministic intensity $\lambda(t; a, b, c, d) = a + b\sin(ct + d)$; and a linear Hawkes process with $\mathcal{H}$-intensity $\lambda^{\mathcal{H}}(t) = (\mu + \alpha \sum \phi(t - t_i))$. The gradient at different values of parameters is computed and depicted in Figure 3.

The result shows three facts. First, every line in the figure touches X-axis at the origin, except for NHPP(c) (indifferentiable). This phenomenon demonstrates that thinning sampling yields asymptotically unbiased parameter estimation, no matter for LSE or MLE. Second, we can see that red and blue lines in the results of first 6 sub-figures overlap significantly, which confirms that thinning gives unbiased gradient estimation for deterministic intensities. Third, in the last two sub-figures, blue lines tend to be on or above the red ones, which demonstrates that thinning makes gradient estimation larger or equal to the ground truth for stochastic intensities.

**Stochastic optimization.** We test thinning sampling for stochastic optimization algorithms, including *SGD* and *Adam*. The task is to learn a linear Hawkes process. We test *Thinning* (p=0.1), sub-interval sampling (*SubInt*), the stochastic optimization learning algorithm (*StoOpt*) proposed in [32], combined with *SGD*, *ADAM* and the typical gradient descent (*GD*). We test on 4 datasets:

- **Synthetic dataset**: We simulate 10 realizations of a 5-dimensional linear Hawkes process, with parameters generated from a uniform distribution. The dataset contains 20k events. We train the model using the entire dataset and the RMSE between the estimated parameters and the ground truth is shown as test error.
- **IPTV dataset** [24]: The dataset consists of IPTV viewing events, which records the timestamps for multiple users watching a video, and the category that the video belongs to. Each user is

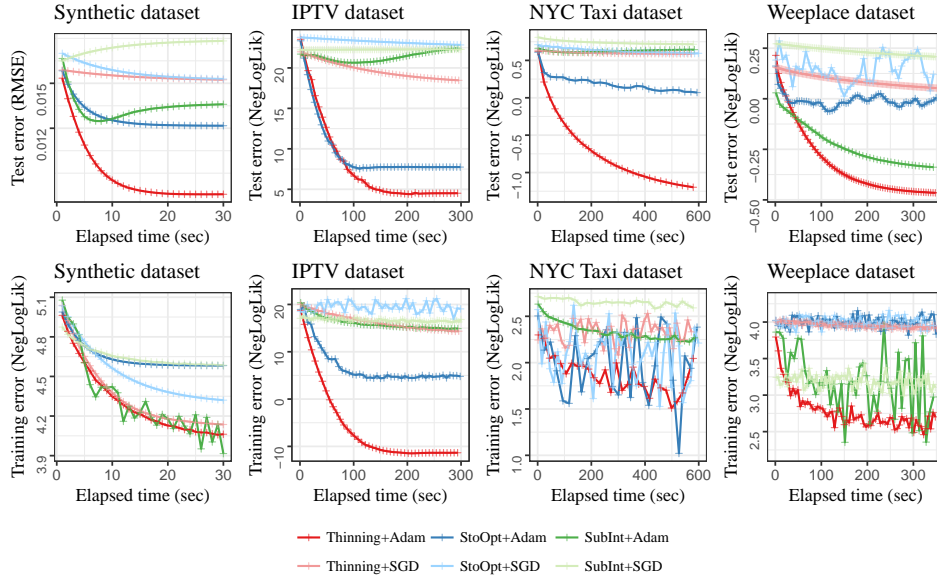

Figure 4: The average convergence curves of different learning algorithms on different datasets.

treated as a realization and each category as a dimension. We select 7 and 3 realizations with 22k and 9k events as training and test datasets, respectively. The number of categories is 16.

- **NYC taxi dataset**: The data is from The New York City Taxi and Limousine Commission[1], which records fields capturing pick-up time, location and payment information of green taxis' orders. We select those trips starting from Manhattan district in the first 10 days of January 2018 and use the 14 areas as dimensions. The training and test datasets contain 60k and 12k events, respectively.
- **Weeplace dataset** [23]: This dataset contains the check-in histories of users at different locations. The categories of events include food, education, outdoors, shops, and 10 others. The check-in histories of 46 and 10 users are selected as training and test dataset, respectively. The sizes of the datasets are 50k and 11k.

We ran each method on each dataset for 10 times. Figure 4 presents the average convergence curves of each method on different datasets. Training of *GD* failed to finish the first iteration given the maximum time shown in Figure 4 for each dataset and thus its results are not presented. From the learning curves, we can see that *Thinning+Adam* outperforms all the competitors in terms of test error on all the datasets. When looking at the *SGD* group alone, *Thinning* also achieves the lowest test error. From the bottom row, we see that *Thinning+Adam* tends to have less fluctuated learning curves. Especially on Weeplace and NYC taxi datasets, the fluctuations of *StoOpt* and *SubInt* are dramatic. This is due to the fact that thinning sampling can better capture the information of the whole timeline, whereas other methods are prone to a zigzag of searching path.

# 8 Conclusion & Discussion

In this paper, we discussed thinning as a downsampling method for point processes. Thinning operation uniformly compresses the intensity on time axis, but its structure is completely preserved. In this way, for parameter estimation, similar performance can be achieved with less input data, as shown in the experiments. We also demonstrated how to estimate gradient on the thinned history, which leads to a novel stochastic optimization algorithm, called TSGD. Experimental results show that TSGD converges faster and has a learning curve with less fluctuations, which can be explained by the theorem that the thinning estimator for gradient has a smaller variance.

In future work, it would be interesting to study other sampling methods, such as Jackknife resampling, for point processes. This work focuses on point processes with decouplable intensities. It will also be interesting to explore a broader assumption to serve more scenarios.

## Acknowledgment

This work is partially supported by the Data Science and Artificial Intelligence Research Centre (DSAIR) and the School of Computer Science and Engineering at Nanyang Technological University.

## Footnotes

[1]https://www1.nyc.gov/site/tlc/index.page

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
