[Supplementary Material · Supplementary.pdf]

# Thinning for Accelerating the Learning of Point Processes (Supplementary Material)

**Lemma 3.2** (Thinned intensities). *Let $\mathcal{F}$ and $\mathcal{G}$ be the full history and thinned history with respect to a p-thinned process $N_p(t)$. Let $\mathcal{H}$ be the internal history of $N(t)$. The following equalities hold:*

*(1)* $\lambda_p^{\mathcal{F}}(t) = p\lambda^{\mathcal{H}}(t)$;

*(2)* $\lambda_p^{\mathcal{G}}(t) = p\mathbb{E}\left[\lambda^{\mathcal{H}}(t)|\mathcal{G}\right]$.

*Proof.* For (1), it can be obtained by taking expectation on both side of $dN_p(t) = B_{N(t)}dN(t)$:

$$\lambda_p^{\mathcal{F}}(t) = \mathbb{E}dN_p(t) = \mathbb{E}B_{N(t)}dN(t) = \lambda^{\mathcal{H}}(t). \tag{1}$$

For (2), Theorem 7.13 in [2] gives a solution to recover the point process given the thinned history be the following conditional expectation:

$$\mathbb{E}\left[N(t)|\mathcal{G}\right] = N_p(t) + \frac{1-p}{p}\int_0^t d\Lambda_p^{\mathcal{G}}(s). \tag{2}$$

Here, $\Lambda_p^{\mathcal{G}}$ is the $\mathcal{G}$-compensator of the p-thinned process, which equals to $\Lambda_p^{\mathcal{G}}(t) = \int_0^t \lambda_p^{\mathcal{G}}(s)ds$. Further,

$$\begin{aligned}
\mathbb{E}\left[\lambda^{\mathcal{H}}(t)|\mathcal{G}\right] &= \lim_{s\to 0}\frac{\mathbb{E}\left[N(t+s)-N(t)|\mathcal{G}\right]}{ds} \\
&= \lim_{s\to 0}\frac{\mathbb{E}\left[N_p(t+s)-N_p(t)|\mathcal{G}\right]}{ds} + \frac{1-p}{p}\lambda_p^{\mathcal{G}}(t) \\
&= \lambda_p^{\mathcal{G}}(t) + \frac{1-p}{p}\lambda_p^{\mathcal{G}}(t) \\
&= \frac{1}{p}\lambda_p^{\mathcal{G}}(t).
\end{aligned}$$

where the desired result follows. $\square$

**Lemma 4.1** (Thinning for parameter estimation of NHPP). *Consider an NHPP $N(t)$ with deterministic intensity $\lambda(t;\theta)$, $t > 0$, $\theta \in \mathbb{R}^d$. If there exists an invertible linear operator $\mathcal{A}: \mathbb{R}^d \to \mathbb{R}^d$ satisfying $\lambda(t;\mathcal{A}\theta) = p\lambda(t;\theta)$, then the M-estimator on thinned history can be written as $\tilde{\theta}_{\mathcal{H}} = \mathcal{A}^{-1}\hat{\theta}_{\mathcal{G}}$ such that $\mathbb{E}\left[\nabla R(\tilde{\theta}_{\mathcal{H}})|\mathcal{G}\right] \xrightarrow{P} 0$, as the number of realizations $n \to \infty$.*

*Proof.* From Theorem 7.13 in [2] we have,

$$\mathbb{E}\big[\nabla R(\tilde{\theta}_{\mathcal{H}})|\mathcal{G}\big] = \mathbb{E}\Big\{\frac{1}{p}\int_0^T H(t;\tilde{\theta}_{\mathcal{H}})\Big[dN_p(t) - p\lambda(t;\tilde{\theta}_{\mathcal{H}})dt\Big|\mathcal{G}\Big]\Big\}$$

$$= \frac{1}{p}\int_0^T H(t;\tilde{\theta}_{\mathcal{H}})\big[dN_p(t) - \lambda(t;\hat{\theta}_{\mathcal{G}})dt\big]$$

$$= \frac{1}{p}\int_0^T H(t;\tilde{\theta}_{\mathcal{H}})\big[dN_p(t) - \lambda(t;\theta_{\mathcal{G}}^*)dt + \lambda(t;\theta_{\mathcal{G}}^*)dt - \lambda(t;\hat{\theta}_{\mathcal{G}})dt\big]$$

$$= \frac{1}{p}\int_0^T H(t;\tilde{\theta}_{\mathcal{H}})\big[\lambda(t;\theta_{\mathcal{G}}^*)dt - \lambda(t;\hat{\theta}_{\mathcal{G}})dt\big] \xrightarrow{P} 0.$$

The last step is due to the asymptotic normality of M-estimator ([1]) that $\hat{\theta}_{\mathcal{G}} \xrightarrow{P} \theta_{\mathcal{G}}^*$ as the number of realizations $n \to \infty$. $\square$

**Theorem 4.3** (Thinning for parameter estimation of decouplable intensities). *Consider a point process* $N(t)$ *with decouplable intensity. If there exist invertible linear operators* $\mathcal{A}$ *and* $\mathcal{B}$ *satisfying* $\mathcal{B}\mathbb{E}\big[m^{\mathcal{H}}(t)|\mathcal{G}\big] = m_p^{\mathcal{G}}(t)$, *where* $m_p^{\mathcal{G}}(t)$ *is the component of thinned intensity* $\lambda_p^{\mathcal{G}}(t)$, *and* $p\mathcal{B}^{-1}g(t;\theta) = g(t;\mathcal{A}\theta)$, *then the M-estimator on thinned history can be written as* $\tilde{\theta}_{\mathcal{H}} = \mathcal{A}^{-1}\hat{\theta}_{\mathcal{G}}$ *such that* $\mathbb{E}\big[\nabla R(\tilde{\theta}_{\mathcal{H}})|\mathcal{G}\big] \xrightarrow{P} 0$, *as the number of realizations* $n \to \infty$. *Particularly, if* $\lambda^{\mathcal{H}}(t;\theta)$ *is linear, then* $\mathcal{A} = p\mathcal{B}^{-1}$.

*Proof.* The proof is similar with the NHPP one. Be definition we have,

$$\mathbb{E}\big[\nabla R(\tilde{\theta}_{\mathcal{H}})|\mathcal{G}\big] = \frac{1}{p}\mathbb{E}\int_0^T \Big\{H^{\mathcal{H}}(t;\tilde{\theta}_{\mathcal{H}})\big[dN_p(t) - p\lambda^{\mathcal{H}}(t;\tilde{\theta}_{\mathcal{H}})dt\big]\Big|\mathcal{G}\Big\}$$

$$= \frac{1}{p}\int_0^T \mathbb{E}\big\{H^{\mathcal{H}}(t;\tilde{\theta}_{\mathcal{H}})|\mathcal{G}\big\}\mathbb{E}\Big\{dN_p(t) - p\lambda^{\mathcal{H}}(t;\tilde{\theta}_{\mathcal{H}})dt\Big|\mathcal{G}\Big\}$$

By the definition of stochastic integral, it suffices to show that $N_p(t) - \int p\lambda^{\mathcal{H}}(t;\tilde{\theta}_{\mathcal{H}})dt$ asymptotically converges to a martingale in probability.

$$\mathbb{E}\Big\{dN_p(t) - p\lambda^{\mathcal{H}}(t;\tilde{\theta}_{\mathcal{H}})dt\Big|\mathcal{G}\Big\} = g(t;\theta^*)^{\mathsf{T}}m^{\mathcal{G}}(t) - pg(t;\tilde{\theta}_{\mathcal{H}})^{\mathsf{T}}\mathbb{E}\big[m^{\mathcal{H}}(t)|\mathcal{G}\big]$$

$$= \big[g(t;\theta^*) - p\mathcal{B}^{-1}g(t;\tilde{\theta}_{\mathcal{H}})\big]^{\mathsf{T}}m^{\mathcal{G}}(t)$$

$$= \big[g(t;\theta^*) - g(t;\mathcal{A}\tilde{\theta}_{\mathcal{H}})\big]^{\mathsf{T}}m^{\mathcal{G}}(t)$$

$$= \big[g(t;\theta^*) - g(t;\hat{\theta}_{\mathcal{G}})\big]^{\mathsf{T}}m^{\mathcal{G}}(t)$$

Since $\hat{\theta}_{\mathcal{G}} \xrightarrow{P} \theta_{\mathcal{G}}^*$ as the number of realizations $n \to \infty$, and $g$ is continuous with respect to $\theta$, $\big[g(t;\theta^*) - g(t;\hat{\theta}_{\mathcal{G}})\big]^{\mathsf{T}}m^{\mathcal{G}}(t) \xrightarrow{P} 0$, which is the desired result. $\square$

**Theorem 5.1** (Thinning for gradient estimation). *Let* $N(t)$ *be a point process with decouplable intensity* $\lambda^{\mathcal{H}}(t;\theta) = g(t;\theta)^{\mathsf{T}}m^{\mathcal{H}}(t)$ *in Eq. (4). If there exist invertible linear operators* $\mathcal{A}$ *and* $\mathcal{B}$ *satisfying* $\mathcal{B}\mathbb{E}\big[m^{\mathcal{H}}(t)|\mathcal{G}\big] = m_p^{\mathcal{G}}(t)$, *where* $m_p^{\mathcal{G}}(t)$ *is the component of thinned intensity* $\lambda_p^{\mathcal{G}}(t)$, *and* $p\mathcal{B}^{-1}g(t;\theta) = g(t;\mathcal{A}\theta)$, *then*

*(1)* $\mathbb{E}\big[\nabla R(\theta)|\mathcal{G}\big] \leqslant 1/p\mathcal{A}^{-1}\nabla R_p(\mathcal{A}\theta)$, *for R is LSE;*

*(2)* $\mathbb{E}\big[\nabla R(\theta)|\mathcal{G}\big] \leqslant \mathcal{A}^{-1}\nabla R_p(\mathcal{A}\theta)$, *for R is MLE.*

*Particularly, if the intensity is deterministic, i.e.,* $m^{\mathcal{H}}(t) = 1$, *both equalities hold.*

*Proof.* By definition of stochastic integral, we have

$$\mathbb{E}\big[\nabla R(\theta)|\mathcal{G}\big] = \mathbb{E}\Big\{\int_0^T H^{\mathcal{H}}(t;\theta)\big[dN(t) - \lambda^{\mathcal{H}}(t;\theta)dt\big]\Big|\mathcal{G}\Big\}$$

$$= \mathbb{E}\Big\{\int_0^T H^{\mathcal{H}}(t;\theta)dN(t)\Big|\mathcal{G}\Big\} - \mathbb{E}\Big\{\int_0^T H^{\mathcal{H}}(t;\theta)\lambda^{\mathcal{H}}(t;\theta)dt\Big|\mathcal{G}\Big\}$$

Here the second term can be bounded by,

$$\mathbb{E}\left\{H^{\mathcal{H}}(t;\theta)\lambda^{\mathcal{H}}(t;\theta)dt\big|\mathcal{G}\right\} \geqslant \mathbb{E}\left\{H^{\mathcal{H}}(t;\theta)\big|\mathcal{G}\right\}\mathbb{E}\left\{\lambda^{\mathcal{H}}(t;\theta)dt\big|\mathcal{G}\right\}$$

According to the definition of forward stochastic integral, the first term can be written as,

$$\mathbb{E}\left\{\int_0^T H^{\mathcal{H}}(t;\theta)dN(t)\big|\mathcal{G}\right\} = \int_0^T \mathbb{E}\left\{H^{\mathcal{H}}(t;\theta)\big|\mathcal{G}\right\}\mathbb{E}\{dN(t)|\mathcal{G}\}$$

Let's look at these components one by one. The condition of the theorem yields,

$$\begin{aligned}
\mathbb{E}\left\{\lambda^{\mathcal{H}}(t;\theta)dt\big|\mathcal{G}\right\} &= g(t;\theta)^{\mathsf{T}}\mathbb{E}\big[m^{\mathcal{H}}(t)|\mathcal{G}\big]dt \\
&= p\mathcal{B}^{-1}g(t;\theta)^{\mathsf{T}}m^{\mathcal{G}}(t)dt \\
&= g(t;\mathcal{A}\theta)^{\mathsf{T}}m^{\mathcal{G}}(t)dt \\
&= \lambda^{\mathcal{G}}(t;\mathcal{A}\theta)dt
\end{aligned} \tag{3}$$

and,

$$\mathbb{E}\{dN(t)|\mathcal{G}\} = \frac{1}{p}dN_p(t). \tag{4}$$

If R is LSE, then we have,

$$\begin{aligned}
\mathbb{E}\left[H^{\mathcal{H}}(t;\theta)|\mathcal{G}\right] &= \nabla\mathbb{E}\left[\lambda^{\mathcal{H}}(t;\theta)|\mathcal{G}\right] \\
&= \nabla_\theta\frac{1}{p}g(t;\mathcal{A}\theta)^{\mathsf{T}}m_p^{\mathcal{G}}(t) \\
&= \mathcal{A}^{-1}\nabla\frac{1}{p}g(t;\mathcal{A}\theta)^{\mathsf{T}}m_p^{\mathcal{G}}(t) \\
&= \mathcal{A}^{-1}H_p^{\mathcal{G}}(t;\mathcal{A}\theta)
\end{aligned} \tag{5}$$

Thus, combining Eq.(3),(4) and (5) yields,

$$\begin{aligned}
\mathbb{E}\big[\nabla R(\theta)|\mathcal{G}\big] &\leqslant \frac{1}{p^2}\mathcal{A}^{-1}\int_0^T H_p^{\mathcal{G}}(t;\mathcal{A}\theta)\big[dN_p(t) - \lambda_p^{\mathcal{G}}(t;\mathcal{A}\theta)dt\big] \\
&= \frac{1}{p}\mathcal{A}^{-1}\nabla R_p(\mathcal{A}\theta).
\end{aligned}$$

If R is MSE, then we have,

$$\begin{aligned}
\mathbb{E}\left[H^{\mathcal{H}}(t;\theta)|\mathcal{G}\right] &= \nabla\mathbb{E}\left[\log\lambda^{\mathcal{H}}(t;\theta)|\mathcal{G}\right] \\
&\geqslant \nabla_\theta\log\left[g(t;\mathcal{A}\theta)^{\mathsf{T}}m_p^{\mathcal{G}}(t)\right] \\
&= \mathcal{A}^{-1}\nabla\log\left[g(t;\mathcal{A}\theta)^{\mathsf{T}}m_p^{\mathcal{G}}(t)\right] \\
&= \mathcal{A}^{-1}H_p^{\mathcal{G}}(t;\mathcal{A}\theta)
\end{aligned} \tag{6}$$

Combining Eq.(3) and Eq.(6) yields the second conclusion,

$$\begin{aligned}
\mathbb{E}\big[\nabla R(\theta)|\mathcal{G}\big] &\leqslant \frac{1}{p}\mathcal{A}^{-1}\int_0^T H_p^{\mathcal{G}}(t;\mathcal{A}\theta)\big[dN_p(t) - \lambda_p^{\mathcal{G}}(t;\mathcal{A}\theta)dt\big] \\
&= \mathcal{A}^{-1}\nabla R_p(\mathcal{A}\theta).
\end{aligned}$$

The proof ends here. $\qquad\square$

**Theorem 5.2** (Variance of gradient estimation). *Let $\nabla\hat{R}^{\mathcal{G}}(\theta)$ and $\nabla R_\ell(\theta)$ be the $p$-thinned and sub-interval gradient at $\theta$, where $\nabla\hat{R}^{\mathcal{G}}(\theta) = 1/p\mathcal{A}^{-1}\nabla R_p(\mathcal{A}\theta)$ for LSE and $\nabla\hat{R}^{\mathcal{G}}(\theta) = \mathcal{A}^{-1}\nabla R_p(\mathcal{A}\theta)$ for MLE. The variance of $p$-thinned gradient is no greater than that of sub-interval gradient:*

$$\mathbb{V}\left[\nabla\hat{R}^{\mathcal{G}}(\theta)\right] \leqslant \mathbb{V}\left[\nabla R_\ell(\theta)\right]. \tag{7}$$

*Proof.* For the RHS, using the law of total variance yields,

$$\mathbb{V}\big[\nabla R_\ell(\theta)\big] = \mathbb{E}\Big\{\mathbb{V}\big[\nabla R_\ell(\theta)|\mathcal{F}\big]\Big\} + \mathbb{V}\Big\{\mathbb{E}\big[\nabla R_\ell(\theta)|\mathcal{F}\big]\Big\}.$$

The first term can be rewritten as,

$$\mathbb{E}\Big\{\mathbb{V}\big[\nabla R_\ell(\theta)|\mathcal{F}\big]\Big\} = \frac{1-p}{p}\mathbb{E}\left\{\int_0^T H^{\mathcal{H}}(t;\theta)\big[dN(t) - \lambda^{\mathcal{H}}(t;\theta)dt\big]\right\}^2$$

$$= \frac{1-p}{p}\mathbb{E}\big[\nabla R(\theta)\big]^2,$$

The second term can be written as,

$$\mathbb{V}\Big\{\mathbb{E}\big[\nabla R_\ell(\theta)|\mathcal{F}\big]\Big\} = \mathbb{V}\big[\nabla R(\theta)\big].$$

Thus, the total variance of $\nabla R_\ell(\theta)$ can be written as,

$$\mathbb{V}\big[\nabla R_\ell(\theta)\big] = \frac{1-p}{p}\mathbb{E}\big[\nabla R(\theta)\big]^2 + \mathbb{V}\big[\nabla R(\theta)\big].$$

Then we consider the LHS, by the definition of variance,

$$\mathbb{V}\big[\nabla\hat{R}^{\mathcal{G}}(\theta)\big] = \mathbb{E}\big[\nabla\hat{R}^{\mathcal{G}}(\theta)\big]^2 - \big[\mathbb{E}\nabla\hat{R}^{\mathcal{G}}(\theta)\big]^2. \tag{8}$$

Apply Theorem **??**, we have,

$$\big[\mathbb{E}\nabla\hat{R}^{\mathcal{G}}(\theta)\big]^2 \geqslant \big[\mathbb{E}\nabla R(\theta)\big]^2. \tag{9}$$

For LSE, since quadratic function is convex, we obtain $\mathbb{E}\big[H_p^{\mathcal{G}}(t;\mathcal{A}\theta)\big]^2 \leqslant \mathbb{E}\big[H^{\mathcal{H}}(t;\theta)\big]^2$. This equivalence also holds for MLE, we omit the proof, since it can be proved similarly. Further, we obtain,

$$\mathbb{E}\big[\nabla\hat{R}^{\mathcal{G}}(\theta)\big]^2 = \frac{1}{p^2}\mathcal{A}^{-1}\mathbb{E}\Big[\int_0^T H_p^{\mathcal{G}}(t;(\mathcal{A}\theta)\big[dN_p(t) - \lambda_p^{\mathcal{G}}(t;\mathcal{A}\theta)dt\big]\Big]^2(\mathcal{A}^{-1})^{\mathsf{T}}$$

$$= \frac{1}{p^2}\mathcal{A}^{-1}\mathbb{E}\Big[\int_0^T H_p^{\mathcal{G}}(t;(\mathcal{A}\theta)\big[dN_p(t) - \lambda_p^{\mathcal{G}}(t;\theta_{\mathcal{G}}^*)dt + \lambda_p^{\mathcal{G}}(t;\theta_{\mathcal{G}}^*)dt - \lambda_p^{\mathcal{G}}(t;\mathcal{A}\theta)dt\big]\Big]^2(\mathcal{A}^{-1})^{\mathsf{T}}$$

$$= \frac{1}{p^2}\mathcal{A}^{-1}\mathbb{E}\left\{\int_0^T H_p^{\mathcal{G}}(t;\mathcal{A}\theta)\big[dN_p(t) - \lambda_p^{\mathcal{G}}(t;\theta_{\mathcal{G}}^*)dt\big]\right\}^2(\mathcal{A}^{-1})^{\mathsf{T}} +$$

$$\frac{1}{p^2}\mathcal{A}^{-1}\mathbb{E}\left\{\int_0^T H_p^{\mathcal{G}}(t;\mathcal{A}\theta)\big[\lambda_p^{\mathcal{G}}(t;\mathcal{A}\theta_{\mathcal{G}}^*) - \lambda_p^{\mathcal{G}}(t;\mathcal{A}\theta)\big]dt\right\}^2(\mathcal{A}^{-1})^{\mathsf{T}} \tag{10}$$

The first term,

$$\frac{1}{p^2}\mathcal{A}^{-1}\mathbb{E}\left\{\int_0^T H_p^{\mathcal{G}}(t;\mathcal{A}\theta)\big[dN_p(t) - \lambda_p^{\mathcal{G}}(t;\theta_{\mathcal{G}}^*)dt\big]\right\}^2(\mathcal{A}^{-1})^{\mathsf{T}}$$

$$= \frac{1}{p^2}\mathcal{A}^{-1}\mathbb{E}\left\{\int_0^T \big[H_p^{\mathcal{G}}(t;\mathcal{A}\theta)\big]^2 dN_p(t)\right\}(\mathcal{A}^{-1})^{\mathsf{T}}$$

$$\leqslant \frac{1}{p^2}\mathbb{E}\left\{\int_0^T \big[H^{\mathcal{H}}(t;\theta)\big]^2 dN_p(t)\right\}$$

$$= \frac{1}{p}\mathbb{E}\int_0^T \big[H^{\mathcal{H}}(t;\theta)\big]^2 dN(t) \tag{11}$$

The second term,

$$\frac{1}{p^2}\mathcal{A}^{-1}\mathbb{E}\left\{\int_0^T H_p^{\mathcal{G}}(t;\mathcal{A}\theta)\big[\lambda_p^{\mathcal{G}}(t;\mathcal{A}\theta_{\mathcal{G}}^*) - \lambda_p^{\mathcal{G}}(t;\mathcal{A}\theta)\big]dt\right\}^2(\mathcal{A}^{-1})^{\mathsf{T}}$$

$$\leqslant \frac{1}{p}\mathbb{E}\left\{\int_0^T H^{\mathcal{H}}(t;\theta)\big[\lambda^{\mathcal{H}}(t;\theta_{\mathcal{H}}^*) - \lambda^{\mathcal{H}}(t;\theta)\big]dt\right\}^2 \tag{12}$$

Substituting Eq. (11) and (12)to Eq. 10 yields

$$\mathbb{E}\left[\nabla\hat{R}^{\mathcal{S}}(\theta)\right]^2 \leqslant \frac{1}{p}\mathbb{E}\int_0^T \left[H^{\mathcal{H}}(t;\theta)\right]^2 dN(t) + \frac{1}{p}\mathbb{E}\left\{\int_0^T H^{\mathcal{H}}(t;\theta)\left[\lambda^{\mathcal{H}}(t;\theta_{\mathcal{H}}^*) - \lambda^{\mathcal{H}}(t;\theta)\right]dt\right\}^2$$

$$= \frac{1}{p}\mathbb{E}\left\{\int_0^T H^{\mathcal{H}}(t;\theta)\left[dN(t) - \lambda^{\mathcal{H}}(t;\theta_{\mathcal{H}}^*)dt\right]\right\}^2$$

$$= \mathbb{E}\left[\nabla R(\theta)\right]^2. \tag{13}$$

Combine Eq.(13) and Eq.(10) to Eq.(8),

$$\mathbb{V}\left[\nabla\hat{R}^{\mathcal{S}}(\theta)\right] \leqslant \frac{1}{p}\mathbb{E}\left[\nabla R(\theta)\right]^2 - \left[\mathbb{E}\nabla R(\theta)\right]^2$$

$$= \frac{1-p}{p}\mathbb{E}\left[\nabla R(\theta)\right]^2 + \mathbb{E}\left[\nabla R(\theta)\right]^2 - \left[\mathbb{E}\nabla R(\theta)\right]^2$$

$$= \frac{1-p}{p}\mathbb{E}\left[\nabla R(\theta)\right]^2 + \mathbb{V}\left[\nabla R(\theta)\right]$$

$$= \mathbb{V}\left[\nabla R_\ell(\theta)\right],$$

which is the desired result. $\square$