[Reviews · NeurIPS 2019]

Reviewer 1



The paper deals with parametric point processes on the real line. The authors show that thinning a point process by keeping each point at random with given probability $p$ is a method that compresses the intensity, but preserves its structure. Hence, it provides a downsampling method. The method seems to be new, even if it is not a major breakthrough. It is more elegant than the different techniques to paste sub-interval downsampling, and the proofs given in the paper are quite general. Yet, it misses an evaluation of the uncertainty of the estimate. The paper is clearly written, even if it is sometimes unnecessarily abstract (see, eg. Definition 2.2 of the stochastic intensity). By way of example, the theoretical results are applied to two particular parametric cases: non-homogeneous Poisson point processes and Hawkes processes. This is a good idea because it helps to understand the general theoretical results, and to see their possible use.

Reviewer 2



The paper rigorously tackles the question of what is the best way to learn point process parameters given the presence of very long sequences which are samples drawn from the point process. There are two ways of training the models by using either sub-intervals or, as recommended in this paper, by thinning the original sequences to make them shorter. The authors first rigorously establish the classes of models for which thinning can work, show that the variance of the gradient of the residue calculated for the thinned sequence is going to be smaller than that calculated over the sub-interval sequence, and show using experimentation that thinning manages to learn the state-of-the-art models. The paper is very well written; each lemma/theorem/definition is followed by a clear example and explanation. The presentation is also smooth and the paper is approachable while remaining rigorous. However, there still are a few parts of the paper which could do with more explanation. Firstly, the authors hint at the gradient for stochastic intensities potentially being unbiased in line 186-190. An example here and potential discussion of the limitations would help contextualize the contribution better, Also, there seems to be a rather sudden jump from learning/modelling one intensity function to multi-variate intensity functions between Definition 4.2 and Experiments. Overall, I believe that the paper makes a significant contribution and should be accepted for publication. Minor: - Line 66: Missing period. - Eq (2): R(.) is undefined. - Line 196: "of applying" - Missing Theorem (reference) in Supplementary, above eqn. 9.

Reviewer 3



The thinning idea of learning point processes is interesting. The paper is well written. The only concern I have is on the applicability of the proposed model. In the real world experiments, only the task to learn a Hawkes process is discussed. However, Hawkes process is a weak baseline and there are many other point process models that are shown to have better performance than Hawkes processes on the IPTV and taxi data. It would improve the paper if these models can be compared and discussed. -------------- thanks the authors for your response, which addressed my concerns. I changed the score accordingly.

Reviewer 4



This paper presents a unique approach to computationally efficient parameter estimation in point process models using thinning properties. The problem is well-motivated, and the background should be clear to those with some familiarity with point process theory. The derivations are thorough and the experiments provide validation to the claims of computational efficiency and accuracy in the case of deterministic intensities. Overall, the goal of accelerating inference for point process models is a practically relevant one for the ML community, and this paper opens the door to further valuable research on computationally efficient, low-variance gradient estimators for more complicated point process models. Though overall this is a good paper, I recommend the following improvements: First, the paper will likely not be digestible to those without significant prior knowledge of point process theory. To broaden the potential audience, I recommend adding some additional background on point processes (perhaps to an appendix; the background in section 2 is quite dense). Second, theorem 4.3 could use additional clarity. The assumption of decouplable intensity functions is clear, but an explanation of how restrictive the assumptions on A and B are in practice would be useful (perhaps add some examples where the assumptions are not satisfied). Third, I think the analysis in section 5 on the bias of the gradient estimators is lacking. The claim in lines 188-189 seems unsubstantiated ("a larger gradient is not a bad thing..."). Please expand upon why we shouldn't be worried about the biases of your estimators when the intensity is not deterministic.

[Author Response · NeurIPS 2019]

We thank all the reviewers for their efforts and constructive comments, which help improve the quality of our paper.

**To Reviewer #1**
**Q1. Lack of the evaluation of the uncertainty of the estimate.** We actually derived an error bound of the empirical
risk. Based on the analysis of the first and second moment of the estimator presented in Theorems 5.1 and 5.2, a
Chebyshev's type error bound can be easily obtained:

$$\mathbb{P}\Big(\big|\nabla\tilde{\mathsf{R}}^{\mathcal{G}} - \mathbb{E}\nabla\tilde{\mathsf{R}}^{\mathcal{G}}(\theta)\big| > \epsilon\Big) \leqslant \frac{\mathbb{V}\big[\nabla\tilde{\mathsf{R}}^{\mathcal{G}}(\theta)\big]}{\epsilon^2} \leqslant \frac{\mathbb{V}\big[\nabla\mathsf{R}_\ell(\theta)\big]}{\epsilon^2} = \frac{\frac{1-p}{p}\mathbb{E}\big[\nabla\mathsf{R}(\theta)\big]^2 + \mathbb{V}\big[\nabla\mathsf{R}(\theta)\big]}{\epsilon^2},$$

for any $\epsilon > 0$. As we reckon this error bound of the gradient estimation is a direct deduction of Theorems 5.1 and 5.2,
we didn't include this result due to page limit. We will present it as a corollary in the final version.
**Q2. Thinning bootstrap procedure.** Bootstrap is not suitable for point processes as it is a resampling method with
replacement, that is, an event could be sampled more than once. This violates the fundamental assumption of point
processes that events do not arrive simultaneously. A point cannot be sampled twice, otherwise the log-likelihood of the
point process will be infinite. Therefore, we propose thinning as a downsampling method, rather than bootstrap as a
resampling method with replacement. Alternatively, Jackknife resampling seems feasible for point processes, and we
plan to explore it in the future. We will add this discussion to the final version.
**Q3. Better figure representing MSE vs p.** Thanks for the suggestion and we will revise our paper accordingly. We
have conducted an experiment on this with the same setting in the paper and will add a new figure (shown below) in the
final version. It shows that RMSE decreases as p increases.

**To Reviewer #2**
**Q1. The definition of R.** R is the likelihood/squared error loss, aka the residue. We will
make this clear in the final version.
**Q2. Make the transition to multi-variate processes a bit smoother.** This is indeed a
helpful suggestion, we will add a discussion to make it better.
**Q3. Discuss the gradient for stochastic intensities calculated on sub-samples potentially**
**being unbiased.** The gradient estimation is unbiased if and only if $\mathbb{E}[\mathcal{H}(t;\theta)\lambda(t;\theta)] =$

$\mathbb{E}\mathcal{H}(t;\theta)\mathbb{E}\lambda(t;\theta)$, as we shown in the proof of Theorem 5.1. However, this result is somehow inaccessible, therefore
we particularly indicate, in the Theorem 5.1, that the gradient estimation for NHPPs are unbiased, but not for all the
stochastic intensities. We also illustrated this result by the experiment shown in Fig. 3 and discussed in Line 266-269,
in order to provide an intuitive understanding. We will add a remark after the theorem to clarify this result.

**To Reviewer #3**
We want to highlight that the aim of this paper is NOT to propose a specific optimization/learning algorithm, but to
answer one of the most fundamental questions in point processes: what is the best sampling method for point processes?
We propose thinning as a solution, and derive important theoretical results of thinning for parameter estimation, gradient
estimation, and stochastic optimization, for point processes with decouplable intensities. In our empirical study, we
applied thinning to learn various state-of-the-art models (MMEL, Granger Causality for Hawkes, and Sparse Low-rank
Hawkes); to estimate the gradient in different types of point processes (NHPP and Hawkes) with different estimators
(MLE and LSE); and also to perform stochastic optimization under different algorithms (SGD and Adam). Through
this study, we show that thinning is a general downsampling solution for point processes with decouplable intensities,
and is not restricted to a specific learning/optimization algorithm or estimator.
**Q1. The applicability of the proposed model. Hawkes is a weak baseline.** The thinning method is applicable to
most, if not all, state-of-the-art models related to parametric point processes, including MMEL, GC, and sparse low-rank
Hawkes, as we shown in the synthetic experiment in Table 1. In the experiment with real datasets for stochastic
optimization, the aim is to test the applicability of thinning to typical and popular optimization methods such as SGD
and Adam. We picked Hawkes as the model to learn as it is the basis of many derivative models. Note that, **Hawkes is**
**not used as a baseline here**; our baselines are other sampling/optimization methods. The main purpose here is NOT
to show that Hawkes is good, but rather to show that thinning is effective when coupled with various optimization
algorithms (SGD and Adam).
**Q2. The focus on decouplable intensities is a bit limited.** As we mentioned in the paper, most state-of-the-art models
of parametric point processes are decouplable (we would be grateful if the reviewer can point out if we have missed any).
Besides, Netcodec (Long Tran, et al., 2015), parametric Hawkes (Liangda Li, et al., 2014), Hawkes with Stochastic
Excitations (Young Lee, et al., 2016) and SLANT (Abir De, et al., 2016) are also decouplable. This general category
covers all NHPPs, compound Poisson processes, renewal processes, marked point processes with independent markers,
and a certain part of doubly stochastic Poisson processes (Cox processes). We believe this is a general assumption with
many interesting properties. We choose this specific class of point processes in this first attempt of the problem as it
facilitates the mathematical preciseness and rigorous theoretical proofs. We agree that a broader assumption will serve
more scenarios, and we will investigate more general classes.

[Meta-Review · NeurIPS 2019]

This paper proposes a method to efficiently learn the parameters of point processes by using thinning to estimate the parameters and gradients. The paper also develops the theory of the bias and variance of the proposed estimators. Finally, the proposed method is validated experimentally. Overall, the reviewers were quite positive about this paper saying it was well written and that the problem is of practical interest to the ML community. Additionally, the actual method and results seem significant and can open up new research areas. The authors seem to have addressed the major concerns of the reviewers in their response. The reviewers have provided the authors with good feedback to make their paper better and the authors should incorporate those suggestions into the camera-ready version.